# Farang (*Psidium guajava* L.) Dried Leaf Extracts: Phytochemical Profiles, Antioxidant, Anti-Diabetic, and Anti-Hemolytic Properties for Ruminant Health and Production

**DOI:** 10.3390/molecules27248987

**Published:** 2022-12-16

**Authors:** Rayudika Aprilia Patindra Purba, Pramote Paengkoum

**Affiliations:** Institute of Agricultural Technology, School of Animal Technology and Innovation, Suranaree University of Technology, Nakhon Ratchasima 30000, Thailand

**Keywords:** liquid chromatography, herbal remedies, plant extracts, phytochemical profiles, ruminant wellness, scavenging, extraction

## Abstract

Due to its advantageous antioxidant phytochemical components, *Psidium guajava* L. has become an indispensable plant in pharmaceutical formulations, playing a crucial role in safeguarding human health. On ruminant animals, however, there has been limited investigation. The purpose of this investigation was to assess the phytochemical profiles and biological potential of Farang (*P. guajava* L.) leaf extracts for ruminant health. Methanolic and hexanoic extracts from various agricultural areas were prepared over a five-month period. By means of HPLC-DAD, vitamin C (ascorbic acid), essential oil (eugenol), tannin (gallic acid), cinnamic acids (caffeic acid, syringic acid, *p*-coumaric acid, sinapic acid, and ferulic acid), and flavonoids (catechin, rutin, myricetin, quercetin, apigenin, and kaempferol) were detected and quantified. Solvent type, but not cultivation site or sampling time, explained the observed variation in phytochemical profile. Phytochemicals were found in lower concentrations in hexanoic extracts than in methanolic extracts. Catechin and sinapic acid were discovered to be the two most abundant phytochemicals in the methanolic extract of Farang leaf, followed by other phenolic compounds, essential oils, and water-soluble vitamins. Compared with the methanolic extract, the hexanoic extract of Farang leaves was less effective at scavenging oxidation in terms of 1,1-diphenyl-2-picrylhydrazyl (DPPH), nitric oxide, and superoxide, and α-glucosidase inhibitory activity. Hexanoic extract was found to be less protective against oxidative damage in ruminant erythrocytes than methanolic extract in terms of inhibiting hemoglobin oxidation, lipid peroxidation, and hemolysis. According to the findings of this study, the leaves of Farang (*P. guajava* L.) are a potential source of phytochemical compounds with wellness properties for ruminant production.

## 1. Introduction

Guava (*Psidium guajava* L.) is a popular fruit consumed fresh, preserved, or processed due to its high nutritional content, particularly in phytochemistry, fiber, and minerals [1]. Guavas have long been utilized as both a food and a medicinal plant in tropical and subtropical regions [2]. Thailand is one of the world’s leading producers of guava fruit. Guava output has expanded significantly in the last decade in response to rising demand for fresh fruit and processed goods. For example, the world demand for guava was approximately 46.5 million metric tons, with Thailand supplying 9.70 million metric tons in 2021 [3]. However, guava fruits that have been processed or consumed yield guava leaves, which are considered waste and are disposed of in a landfill. It is expected to increase by more than three times by 2050 [3]. Waste usage studies have established that guava leaves are a rich source of several health-promoting micro- and macronutrients, as well as bioactive substances [1,2,3,4]. Guava leaf can therefore be viewed as a significant resource for the creation of a number of value-added goods due to its accessibility and inherent nutritional content.

Phytochemicals derived from guava or Farang leaves (*P. guajava* L.) have long been recognized in Thailand to provide health benefits for humans [1,2], but their reputation was recently boosted following a number of encouraging clinical studies in domesticated animals in a variety of disease profiles that appear to confirm efficacy. One of these health-promoting characteristics is the reduction in oxidative stress, which is a well-known modern farm method in Thailand for improving animal performance and health [5,6,7]. The presence of reactive oxygen species within the cell causes oxidative stress because they can overcome the cell’s own innate antioxidant defenses. Oxidative stress may be involved in a range of clinical problems in domesticated animals, such as ruminants, including those linked to animal production and general health [5]. The diagnosis of oxidative stress frequently requires the utilization of analytical methods that are specialized, given that there are frequently no obvious indicators of the condition. The dynamic balance of pro-oxidants and antioxidants in the blood is typically sufficiently explained by a single test of total antioxidant capacity, which provides useful data. It is generally believed that reactive oxygen species metabolites are created during normal metabolism and that pro-oxidants derived from plant phytochemicals (vitamins, essential oils, tannins, flavonoids, cinnamic acid, etc.) can enter that metabolism, thereby exerting a limited influence on reactive oxygen species metabolites [7,8,9]. Mastitis, lipid peroxidation (hepatic metabolism), insulin resistance (which has been extensively studied in relation to type II diabetes mellitus), and lipid metabolism in cows during early postpartum or late lactation have all been linked to oxidative stress [5,10,11]. Despite the fact that oxidative stress has been related to these numerous diseases, there is a great deal to learn about its role in ruminant health and production, such as the effect of phytochemicals derived from Farang leaves.

Few studies have been conducted on the chemical composition and biological properties of leaves [12,13,14], and to our knowledge, there were few data on the effects of Farang (*P. guajava*) leaves on ruminant animal models [15,16]. Therefore, the goal of this study was to assess the phytochemical profiles of Farang leaf extracts using high performance liquid chromatography with diode array detection (HPLC-DAD) and their biological potential for ruminant health. In the presence of the radicals 1,1-diphenyl-2-picrylhydrazyl (DPPH^•^), nitric oxide (^•^NO), and superoxide (O_2_^•−^), the antioxidant activity was assessed. Farang leaves were also tested for their α-glucosidase inhibitory potential and protection against induced oxidative damage in ruminant erythrocytes in term of hemoglobin oxidation, lipid peroxidation, and hemolysis. Moreover, because nutritional antioxidants may be linked to some of the purported positive benefits of the crude leaf matrix [14,17] and environmental stresses may be triggered to significantly contribute to differences in the relative composition of a multiplicity of phytometabolites and biosynthesized in plants [18], this study was carried out to isolate, characterize, and quantify the chemical composition of two methanolic and hexanoic extracts derived from various agricultural areas over a five-month harvesting period.

## 2. Results

### 2.1. Phytochemistry Profiling

For the extraction of phytochemical components from Farang leaves, the use of polar and non-polar solvents, such as methanol and hexanol, respectively, seemed to be a viable choice. Without the need for time-consuming fractionation, HPLC-DAD was able to identify the following 14 phytochemical components: vitamin C (ascorbic acid), essential oil (eugenol), tannin (gallic acid), cinnamic acids (caffeic acid, syringic acid, *p*-coumaric acid, sinapic acid, and ferulic acid), and flavonoids (catechin, rutin, myricetin, quercetin, apigenin, and kaempferol). Nonetheless, 39 min after the commencement of the chromatographic run, the hexanoic plant extract exhibited an indistinct peak and poor resolution for these phytochemical components (Figure 1). Hexanoic plant extracts can recover the essential oil, tannin, cinnamic acid, and flavonoid content of Farang leaves, but not ascorbic acid, or sinapic acid. Methanolic plant extracts, on the other hand, can compensate for their lack of ascorbic acid (0.19 mg/g), and sinapic acid by containing approximately 35.24% total cinnamic acids. The differences between methanolic and hexanoic plant extracts were most pronounced for catechin and *p*-coumaric acid, which were 214 times higher and 9 times higher in methanolic plant extracts than hexanoic plant extracts, respectively (Table 1). Table 2 lists the amounts of total tannin, flavonoid, cinnamic acid, essential oil, and vitamin C content in Farang leaves that were harvested over a five-month period from various cultivation sites. The cultivation site or sampling period had no impact on the amounts of tannin, flavonoid, cinnamic acid, essential oil, or vitamin C in Farang leaves.

### 2.2. Antioxidant Activity

Antioxidant activity of two methanolic and hexanoic plant extracts were determined in a concentration-dependent manner in the scavenging system of the radicals DPPH^•^, ^•^NO, and O_2_^•−^ (Figure 2). The IC_50_ values of those plant extracts were summarized in Table 3. Compared with hexanoic plant extracts, methanolic plant extracts had a greater capacity to scavenge DPPH^•^, ^•^NO, and O_2_^•−^ (*p* < 0.05).

### 2.3. α-Glucosidase Inhibitory Activity

The concentration-dependent α-glucosidase inhibitory potential of two methanolic and hexanoic plant extracts was determined (Figure 2D). The methanolic extract’s minimum in vitro α-glucosidase inhibitory potential was 44.17% at 10 mg/mL, and the maximum in vitro activity was 99.52% at 500 mg/mL. The minimum in vitro α-glucosidase inhibitory activity of the hexanoic extract was 15.49% at a dose of 10 mg/mL, while its maximum in vitro activity was 72.80% at a concentration of 500 mg/mL. The IC_50_ values of those plant extracts are presented in Table 3. Methanolic plant extracts had a greater α-glucosidase inhibitory activity than hexanoic plant extracts (*p* < 0.05).

### 2.4. Evaluation of Farang Leaf Extracts on Oxidative Damage in Ruminant Blood Erythrocytes

In vitro ROO^•^-induced oxidative damage in ruminant erythrocytes was evaluated for inhibition of hemoglobin oxidation, lipid peroxidation, and hemolysis by two methanolic and hexanoic plant extracts of Farang (*P. guajava* L.) leaves in a concentration-dependent manner (Figure 3). The IC_50_ values of these plant extracts are presented in Table 3. Hexanoic plant extracts were less effective than methanolic plant extracts at inhibiting hemoglobin oxidation, lipid peroxidation, and hemolysis (*p* < 0.05).

## 3. Discussion

To the best of the author’s knowledge, the current findings are the first to report on the presence of phytochemical profiling in Farang (*P. guajava* L.) leaves, and the data indicated that the positive effects of its antioxidant phytochemical components, which are a part of the body’s antioxidant defense system and play a significant role in safeguarding human or ruminant health not only because of vitamins, tannins, and flavonoids but also because of the presence of essential oils and cinnamic acids. On the other hand, the relative abundance of vitamin C (ascorbic acid), tannin (gallic acid), cinnamic acids (caffeic acid, *p*-coumaric acid, and ferulic acid), and flavonoids (catechin, rutin, myricetin, quercetin, apigenin, and kaempferol) were similar to previous reports for tropical and subtropical countries [1,13,19]. Furthermore, our findings suggested that the similarities in the relative composition of the many phytometabolites and biosynthetic products found in the medicinal leaves of the Farang plant could not be attributed to any significant change in the environment. Similarly, previous research found similarities in the phytochemical profiles of Thai medicinal herbs, despite the fact that the plants were collected from different locations and at different times [14,17]. Thus, the overall effect of different cultivation sites and harvesting times of Farang leaves can be deemed negligible.

In this study, we used standard pharmaceutical procedures and a low-cost Soxhlet apparatus. Soxhlet extraction is a sophisticated method for studying the chemical composition of medicinal plants [17]. Both the extraction process and the solvent used in reversed-phase liquid chromatography (HPLC research) influenced extraction yield and susceptibility [14]. This was consistent with the current findings, which showed that hexanoic plant extracts reduced the resolution of the separated peak area and the consistency of the observed retention length when compared with methanolic plant extracts. One possible explanation is that hexane has little effect on the recovery of phytochemicals from polar plant matrices. In the presence of non-polar molecules, hexane, which has a higher boiling point than methanol, is widely used as an organic eluent difficult to elute the phytochemicals. Methanol, on the other hand, has a lower boiling point than hexane and is frequently used as an organic eluent in the presence of polar and non-polar molecules that easily elute phytochemicals. Thus, the lower boiling point and broad polarity of methanol may explain why it is an effective attractant for the specific bioactive compounds of Farang leaves, allowing for easier elution of the phytochemicals into methanol rather than hexane [14]. Another possibility is that polar chemicals in the Farang plant matrices being transported through the column are more firmly absorbed by the polar silica than nonpolar compounds, which pass more quickly through the column and are eluted faster than the polar ones [20].

Reactive oxygen species are unavoidably created in ruminants as a typical consequence of the metabolic process of the cell. These species include the superoxide anion, the nitric oxide radical, and the peroxyl radical (ROO^•^). Particular examples of these radicals are the superoxide anion and the nitric oxide radical, both of which have been related to the development of a wide spectrum of pathophysiologic diseases. The hydroxyl radical is a potently reactive oxidizing species that is created when superoxide combines with specific transition metal ions. The existence of these radicals in the bodies of long-lived ruminants led to the evolution of endogenous systems capable of reducing their levels [7,8]. On the other hand, when these defense systems fail or are insufficient, antioxidants in the diet become vitally important as possible protective agents [6,21,22]. Here, our results suggest that Farang leaves may have health-promoting properties in regulating reactive oxygen species since they contain an abundance of promising antioxidant phytochemicals, ordered as follows: catechin > sinapic acid > *p*-coumaric acid > apigenin > ferulic acid > gallic acid > caffeic acid > eugenol > rutin > myricetin > quercetin > ascorbic acid > kaempferol > syringic acid. Instead of flavonoids, the inclusion of other phytochemical compounds in the extracts cannot be ruled out; higher quantities of tannin, essential oils, and cinnamic acids in the extracts may be connected to their increased antioxidant activities.

The DPPH^•^, nitric oxide, and superoxide tests are three commonly concurrent techniques through which antioxidants neutralize free radicals [23]. In the current study, the antioxidant activity of methanolic or hexanoic extracts of Farang leaf against DPPH^•^, superoxide anion, and nitric oxide radicals was comparable with previous results [1,13,19]. Nonetheless, our findings consistently demonstrated that hexanoic extracts of Farang leaves were less effective against DPPH^•^, superoxide anion, and nitric oxide radicals than methanolic extracts. This occurrence can be attributed to the existence of a variety of promising antioxidant phytochemical components in extracts, which vary depending on the extraction method used. For example, the lack of ascorbic acid and sinapic acid in hexanoic extract can explain why it is weaker than methanolic extract in preventing the beginning of free radical-mediated chain reactions by stabilizing reactive species before they can engage in harmful processes. There may be more evidence for this notion in the hexanoic extract’s relatively low levels of eugenol, an essential oil, and the methanolic extract’s relatively high levels of catechin, a flavonoid. Despite many previous observations concluding that catechin is the most abundant phytochemical component in guava leaves to increase antioxidant capacity [1,13,19,24], our study revealed that catechin is not the only antioxidant-promoting compound in guava leaves; ascorbic acid, sinapic acid, and essential oil have also been identified as key components of the leaves.

Numerous studies have been carried out to investigate the effect of guava leaf extracts on diabetic activity. Recent systematic research indicates that guava leaf aqueous and ethanolic extracts, but not methanolic extracts, significantly reduced blood sugar levels in humans [1] and rats [13,19]. In terms of ruminant glucosidase inhibition, the current study provided a preliminary report on the physiologically active constituent(s) and their derivatives discovered in Farang leaves extracted with methanol and hexanoic extracts. Methanolic plant extract inhibited α-glucosidase more effectively than hexanoic plant extract, which can be attributed to the presence of more cinnamic acids and flavonoids. The same factor that made methanol preferable for extracting Farang leaf phytochemical constituents is likely to be at work here. According to our findings, the presence of a hydroxyl group in cinnamic acid, as well as a possible synergistic interaction with a flavonoid, is important for producing significant inhibition of α-glucosidase. Cinnamic acids, particularly caffeic acid, ferulic acid, and sinapic acid, are widely regarded to inhibit intestinal α-glucosidase, which is one of the therapeutic strategies [25]. More recently, a study on high-fat diet-fed rats lowered plasma glucose concentrations found that supplementation with pure sinapic acid mixed with flavonoids significantly reduced plasma glucose levels in these high-fat rats [26]. Flavonoid structure, location, and the number of hydroxyl groups can all explain why the desired effect was influenced [27]. Quercetin, for example, is a powerful inhibitor of flavonoids. Quercetin, on the other hand, interacted with other flavonoids, most notably catechin and myricetin [28]. These synergistic flavonoid members appeared to promote α-glucosidase-flavonoid interaction, whereas hydroxyl group metalocation decreased band I electron cloud density, resulting in less inhibitory activity. Based on previous research with plant-derived cinnamic acid or flavonoid derivatives [1,25,29], it is proposed that cinnamic acid or flavonoid derivatives in Farang leaves may contribute to a mechanism in hyperglycemia control by inhibiting α-glucosidase, resulting in a decrease in hemoglobin A1c (HbA1c). Lowering HbA1c levels in diabetic ruminants may reduce the occurrence of chronic vascular complications or other possibly deteriorating nutrigenomic points. Unfortunately, neither pure cinnamic acid nor fragmented cinnamic acid nor flavonoids from Farang leaves were evaluated independently in the present in vitro study; hence, future research is required to throw light on this issue.

In ruminants, erythrocytes are the most prevalent form of circulating cell, and they are primarily responsible for gas exchange during respiration. In addition to this, these cells take part in a variety of immunologically intricate processes (antibodies, complement, and bacteria; [30]). Their membranes, on the other hand, are high in polyunsaturated fatty acids. Because of this, as well as the fact that they transport oxygen, they are perfect targets for free radicals and possible promoters of reactive oxygen species. The production of alkyl radicals in the presence of oxygen can result in hemoglobin oxidation, lipid peroxidation, and finally hemolysis [31]. Scientific investigations had connected the long-term accumulation of reactive oxygen species induced by a high temperature humidity index to lower ruminant animal productivity [32,33] and immune function in tropical conditions [6,7,34]. Within the scope of this massive issue, we investigated the capacity of methanolic and hexanoic leaf extracts to inhibit the peroxyl radical formation, the reactive oxygen species responsible for activating hemoglobin oxidation and generating methemoglobin. For the first time, we report the results of an inhibitory experiment for hemoglobin oxidation that was conducted using extracts from Farang leaves.

Methemoglobin is formed through the oxidation of hemoglobin when the iron in the heme group is not in its normal state. This causes oxidative stress, the breakdown of lipids, and the alteration of protein interactions, all of which impair the equilibrium and resilience of the erythrocyte membrane [35]. In addition, the random use of high-fat, direct-fed dairy goats in the current investigation can indicate that all animals used were in a state of elevated peroxidation. Thus, at least partially, oxidative stress events were anticipated and terminated in the erythrocytes of goats as a result of an increase in reactive oxygen species caused by a high temperature humidity index and a high-fat diet. The most generally used method for assessing how much lipid peroxidation has occurred in biomaterials is the TBARS test [7,31]. Guava leaf methanolic extract had been shown to reduce lipid peroxidation in rat plasma [36]. The current study, however, is the first of its kind to see if Farang leaves can reduce lipid peroxidation in ruminant erythrocytes.

In accordance with previous research [4,37,38], our results indicated that ascorbic acid, eugenol, gallic acid, cinnamic acids (caffeic acid, syringic acid, *p*-coumaric acid, sinapic acid, and ferulic acid), and flavonoids (catechin, rutin, myricetin, quercetin, apigenin, and kaempferol) and their derivatives have hydroxyl group substitutions that are associated with their protective potential. Because of its chemical composition and well-known liposolubility, flavonoids, for example, were shown to provide anti-hemolytic protection (the C2=C3 connection of flavonoids’ C ring increased antioxidant capacity [39]). This made it possible for them to absorb into the membrane, where they then functioned as antioxidants to prevent damage to the membrane by eliminating potentially harmful species. According to the findings of Rjeibi, et al. [40], eugenol derived from plants has the potential to prevent H_2_O_2_-induced hemolysis in human erythrocytes. Our findings were also consistent with prior research [41], which discovered that plant cinnamic acid, which included caffeic, *p*-coumaric, syringic, sinapic acid, cinnamic, and ferulic acid regulated human erythrocyte suspension and hypotonicity-induced hemolysis. Indeed, the phytochemical composition of Farang leaves was discovered in significantly higher amounts in methanolic plant extracts, and more hydroxyl groups acting as antioxidant agents in ruminant circulation cells may be expected.

In this study, methanolic plant extract inhibited hemoglobin, lipid peroxidation, and hemolysis more efficiently than hexanoic plant extract, which can be attributable to the availability of more antioxidant phytochemical components, as discussed previously. The significant protective effect of natural antioxidants against the destruction of erythrocyte membranes by existing reactive oxygen species in ruminants can be related to these potential phytochemical components. Indeed, as measured in ruminant erythrocytes, our findings showed a moderate improvement in the potential for Farang leaf phytochemicals to serve as a viable alternative to natural antioxidants. We observed that this investigation was proposed as a preliminary dietary approach for ruminant health and production in tropical environments. However, empirical research into other types of liquid blood, such as plasma, as well as gene expression and oxidative indicators in ruminal fluids, can provide support for this hypothesis. This may be somewhat attributable to the limits of our investigation. Another disadvantage is the lack of further measures to support the rumen fermentation data. Indeed, flavonoid, tannin, or essential oil can modify rumen microbial environments, which has ramifications for ruminant lifetime performance and welfare. This is in line with earlier studies [7,42] that found significant effects on decreased levels of oxidative stress markers in physiological tissues such as the mammary gland and physiological fluids such as blood, ruminal fluid, and milk of early lactation goats fed *Piper betle* leaves, which contain flavonoids, essential oils, and phenolic acids. These goats also underwent healthy digestion of their feed and fermentation, both of which are crucial to maintaining health.

## 4. Materials and Methods

### 4.1. Standards and Reagents

Glacial acetic acid, acetonitrile, methanol, and hexane, which were employed in the extraction procedure, were acquired from Labscan (Bangkok, Thailand); the solvents’ purities were above 99%. Acarbose, gallic acid, catechin, rutin, myricetin, quercetin, apigenin, kaempferol, caffeic acid, syringic acid, *p*-coumaric acid, ferulic acid, and sinapic acid, as well as eugenol, were all acquired from Sigma Chemical (purity: >99%, St. Louis, MO, USA). The vitamin C standard (ascorbic acid) was acquired from Carlo Erba (purity: >99%, Strada Rivoltana, France). A Milli-Q water purification system was used to prepare the water for preparation, extraction, and liquid chromatography (Millipore, Illkirch-Graffenstaden, France). The following chemicals were purchased from Sigma-Aldrich (St. Louis, MO, USA): 1,1-diphenyl-2-picrylhydrazyl (DPPH^•^), β-nicotinamide adenine dinucleotide (NADH), phenazine methosulfate (PMS), nitrotetrazolium blue chloride (NBT), α-glucosidase from Saccharomyces cerevisiae (type I, lyophilized powder), phosphate-buffered saline (PBS), trypan blue and 2,2′-azobis (2-ethylpropionamidine) dihydrochloride (AAPH), thiobarbituric acid (TBA), trichloroacetic acid (TCA), and tert-butyl hydroperoxide (t-BHP). The following chemicals were purchased from Merck KGaA (Darmstadt, Germany): N-(1-naphthyl) ethylenediamine dihydrochloride, sulfanilamide, 4-nitrophenyl-alpha-D-glucopyranoside (pNPG), and sodium nitroprusside dihydrate (SNP).

### 4.2. Leave Sample, Extraction and Phytochemistry Profiling

The operation for collecting leafy materials was carried out using previous approaches [14,43]. Two cultivation sites, cultivation site I (CS1) and cultivation site II, were used to segregate plant samples (CS2). CS1 was carried out at the Suranaree University of Technology Organic Farm (14°87′1593″ N, 102°02′5890″ E), and CS2 at the temporary garden near the Goat and Sheep Research Center (14°88′532″ N, 102°00′4633″ E), both in Nakhon Ratchasima (northeastern Thailand, at an elevation of 243 m above sea level). CS1 and CS2 samples were gathered from fresh sources for five months in a row (August–December). Following the separation of the flowers and fruits (if any), the leaves were rinsed, blanched with steam at 90 °C for 1 min, and stored at 20 °C until further examination.

The samples CS1 and CS2 were freeze-dried using a lyophilizer (GAMMA 2–16 LSC, CHRiST, Osterode am Harz, Germany). The plant samples underwent a 24-h process of being frozen at −80 °C and vacuum-dried at −15 °C in the condenser. The lyophilized leaves were homogenized and powdered in a Retsch mill with a mesh size of 1 mm (Retsch SM 100 mill, Haan, Germany). Powdered leaves were stored in plastic bags in a vacuum desiccator at 25 °C and 34% humidity until usage.

Vitamin C (ascorbic acid), essential oil (eugenol), tannin (gallic acid), cinnamic acids (caffeic acid, syringic acid, *p*-coumaric acid, sinapic acid, and ferulic acid), and flavonoids (catechin, rutin, myricetin, quercetin, apigenin, and kaempferol) evaluated individually in methanol and hexane were extracted utilizing a Soxhlet extraction system, as described previously [14,17]. Briefly, plant extracts were prepared by extracting 5 g of dried leaf powder with 20 mL of methanol for 3–4 h using the Soxhlet apparatus. All plant extracts were combined after these Soxhlet procedures were completed three times. The plant extract was then evaporated with a Rotavapor (Buchi R300, Flawil, Switzerland). The plant extract was decanted into a volumetric flask after being filtered with a 0.45 µm polyvinylidene difluoride (PVDF) syringe filter (Merck, Germany). In order to prepare the recovered plant extract for further liquid chromatography analysis, the volume was raised to 10 mL using the proper solvents and stored at 20 °C. The hexane extract of the plant was extracted using the same methods as the methanol extract.

On the working day, the obtained plant extracts were evaporated to obtain dry materials. One milligram of evaporated plant extract was mixed in 0.5 mL of mobile phase solution (1:9, HPLC-grade acetonitrile: 1% acetic acid), vortexed, and filtered through a 0.45 µm PVDF syringe filter. The standard stock solution, calibration standard, quality control sample, chromatographic conditions, and computation were all prepared according to a predetermined protocol [14,44]. Twenty injections of each obtained extract (methanolic or hexanoic) from leaves were equipped and analyzed for 65 min using a reversed-phase Zorbax SB-C18 column (3.5 m particle size, i.d., 4.6 mm × 250 mm, Agilent Technologies, Santa Clara, CA, USA). The HPLC setup included a DAD (61315D), a 10 mm flow cell, four quaternary pumps (61311B) for the solvent supply system, and an automated sample injection valve with a 100 µL loop. Data integrity and chromatographic data analysis were carried out using an Agilent OpenLAB CDS 1.8.1 system manager. The mobile phase contained 1% acetic acid and HPLC grade acetonitrile (1:9). A flow rate of 0.9 mL/min was employed with a binary gradient of (A) acetonitrile and (B) 1% acetic acid to accomplish chromatographic separation. The entire gradient elution mechanism was described in the preceding technique [14]. A photodiode array UV detector set at 272, 280, and 310 nm and was used to evaluate the absorption of the chemicals tested and measured. Fourteen external standards, including ascorbic acid, gallic acid, catechin, rutin, myricetin, quercetin, apigenin, kaempferol, as well as caffeic acid, syringic acid, *p*-coumaric acid, ferulic acid, and sinapic acid, were efficiently synthesized, compounded, and examined in 65 min. The peak area was used for quantification (dilution included), with a respective external standard calibration curve used in accordance with the previous approach [14,44]. Values were provided as mg/g on a dry weight basis (Table 1).

### 4.3. Assay Preparations for Antioxidant Activity

Using a previously known method [45], the antioxidant activity of methanolic and hexanoic extracts of Farang leaves against the radicals DPPH^•^, nitric oxide (^•^NO) and superoxide (O_2_^•−^) was evaluated. As a positive control, ascorbic acid was used. Each experiment was carried out six times, with the findings provided as IC_50_ values (mg/mL). For instance, in DPPH^•^ assay, all components of the samples were previously redissolved in methanol (25 μL) and placed in the different wells of the microplate, followed by the addition of 200 μL of 150 mM methanolic DPPH. For each extract, 16 different dilutions were prepared, placed into a 96-well plate, and read at 515 nm. In O_2_^•−^ assay, all components of the samples were dissolved in phosphate buffer (19 mM, pH 7.4). For each extract, 16 different dilutions (10–500 mg/mL) were prepared, placed into a 96-well plate, tested using NBT, and monitored at 562 nm. In ^•^NO assay, all components of the samples were dissolved in phosphate buffer (100 mM, pH 7.4). For each extract, 16 different dilutions were prepared and placed into a 96-well plate. The chromophore formed with Griess reagent was read at 562 nm.

### 4.4. Assay Preparation for α-Glucosidase Inhibitory Activity

In order to evaluate whether or not α-glucosidase activity was inhibited, the Ellman method was applied, as was performed in the earlier description [46]. As a positive control, acarbose was used. Each experiment was carried out six times. A 96-well plate was used to test sixteen different concentrations. In each well, 50 μL of Farang leaf extract dissolved in potassium phosphate buffer, 150 μL of potassium phosphate buffer, and 100 μL of 4-nitrophenyl-α-D-glucopyranoside were added (PNP-G). The reaction was started by adding 25 μL of enzyme, and after incubation the absorbance at 405 nm was measured.

### 4.5. Assay Preparation for In Vitro ROO^•^-Induced Oxidative Damage in Ruminant Erythrocytes

In order to examine the in vitro ROO^•^-induced oxidative damage in ruminant erythrocytes in terms of inhibition of hemoglobin oxidation, lipid peroxidation, and hemolysis, one milligram of evaporated plant extract (methanolic or hexanoic) was dissolved in one milliliter of PBS and diluted sixteen times. Quercetin was utilized as a positive control. Each experiment was repeated six times, with the results reported as IC_50_ values (mg/mL). Blood samples (5 mL) were withdrawn from the jugular veins of random high-fat diet-fed dairy goats on the university farm and placed in evacuated tubes containing K_3_EDTA. Feeding management and animal ethics issues for used dairy goats had been addressed [47,48,49]. Erythrocytes were extracted using a previously reported technique [8,47].

The inhibition of hemoglobin (Hb) oxidation was measured by the ability of Farang extracts to reduce methemoglobin production following to previous report [31]. Methemoglobin is produced when oxyhemoglobin combines with AAPH at a temperature of 38 °C in a water bath. This results in the disintegration of AAPH, which is dissolved in PBS. The reaction mixture was formed by combining 100 μL of previously prepared PBS-fixed extract with 200 μL of erythrocyte solution (1250 × 10^6^ cells/mL, final density). The control and blank were conducted by substituting 100 μL of PBS for the sample. The reaction mixtures were incubated for 30 min in a water bath at 38 °C with slow agitation (50 rpm). Following incubation, 200 μL of AAPH (50 mM final concentration) was added to the mixture (except in the blank), followed by 4 h of incubation under the identical conditions described previously. The entire volume (500 μL) was transferred to 1.5 mL conic Eppendorf tubes and centrifuged for 6 min at 4 °C at 1500× g. The supernatant (300 μL) was deposited in a 96-well plate, and the absorbance at 630 nm was measured.

The production of thiobarbituric acid-reactive compounds was used to indirectly measure lipid peroxidation in erythrocytes (TBARS) in accordance with a prior publication [31]. Previous PBS-fixed extract was combined with a suspension of goat cells (500 × 10^6^ cells/mL, final density) at 38 °C for 30 min with mild agitation (50 rpm). Following incubation, *tert*-butyl hydroperoxide (tBHP) (0.2 mM final concentration) was added to the medium, which was then incubated for 30 min at 38 °C with mild agitation. After incubation, the entire contents were collected and transferred to a 1.5 mL conical Eppendorf tube, and 28% (*w*/*v*) trichloroacetic acid (TCA) was added to induce protein precipitation, followed by a 10 min centrifugation at 16,000× *g* at 18 °C. To produce TBARS from malondialdehyde (MDA) and thiobarbituric acid (TBA), the supernatant was placed in a 2 mL conical test tube (with screw cover), followed by the addition of TBA 1% (*w*/*v*). The resultant mixture was then placed in a water bath and heated at 38 °C for 15 min. After waiting for the test tubes to reach room temperature, an absorbance reading was taken at 532 nm.

AAPH heat breakdown causes ROO^•^ generation and lysis events. The previous methodology [31] was utilized to test goat erythrocyte hemolysis by measuring Hb release after membrane breaking produced by the hemolytic process. Previous PBS-fixed extract (100 μL) was mixed with 200 μL of goat erythrocyte suspension (1775 × 10^6^ cells/mL) and incubated in a water bath at 38 °C for 30 min with mild agitation (50 rpm). Following the incubation period, 200 μL of AAPH (17 mM, final concentration) was added to the mixture (except in the blank), followed by a 3-h incubation period under the same conditions. The entire amount (500 μL) was poured into a 1.5 mL conic-Eppendorf tube and centrifuged for 5 min at 4 °C at 1500× *g*. The absorbance at 540 nm was determined after transferring 300 μL of supernatant to a 96-well plate.

### 4.6. Statistical Analysis

Phytochemistry profiling on total vitamin, essential oil, tannin, cinnamic acid, and flavonoid contents in leafy extracts growing in two sites and sampled at five different times were computed using GraphPad Prism 9.0 and a completely randomized design with repeated measures (GraphPad Software, Inc., San Diego, CA, USA). The covariance structure of the compound symmetry was fitted using the mixed effects model of GraphPad Prism’s Akaike’s information criterion. The Kolmogorov–Smirnov test was used to confirm that the data had a normal distribution. Antioxidant, anti-diabetic, and anti-hemolytic activities were evaluated using the Student’s test. The least-squares means were reported, and significance was assessed using Tukey’s honestly significant difference (HSD) at a level of *p* > 0.05.

## 5. Conclusions

We discovered an abundance of promising antioxidant phytochemicals in Farang (*Psidium guajava* L.) leaves, which are listed the following order: catechin > sinapic acid > *p*-coumaric acid > apigenin > ferulic acid > gallic acid > caffeic acid > eugenol > rutin > myricetin > quercetin > ascorbic acid > kaempferol > syringic acid. The solvent type, cultivated site, and time sampling all played a role in the variances in those phytochemistry profiles. Remarkably, the solvent utilized had no effect on the physicochemical profiles of leaf extracts, nor did the location of the plant or the time of collection. Our findings also validated the notion that the solvent employed inhibited the activities of DPPH, nitric oxide, superoxide, and the enzyme α-glucosidase, as well as modified the inhibition of ruminant erythrocyte hemoglobin oxidation, lipid peroxidation, and hemolysis. These benefits can be ascribed to the leaves of the Farang plant, which are a possible source of phytochemical substances with qualities that are advantageous to the health of ruminant production. If validated by at least in vitro research on nutritional digestion and fermentation, these findings have the potential to open new routes in the hunt for natural antioxidants and antimicrobials that can be investigated in future in vivo investigations.

## Figures and Tables

**Figure 1 molecules-27-08987-f001:**
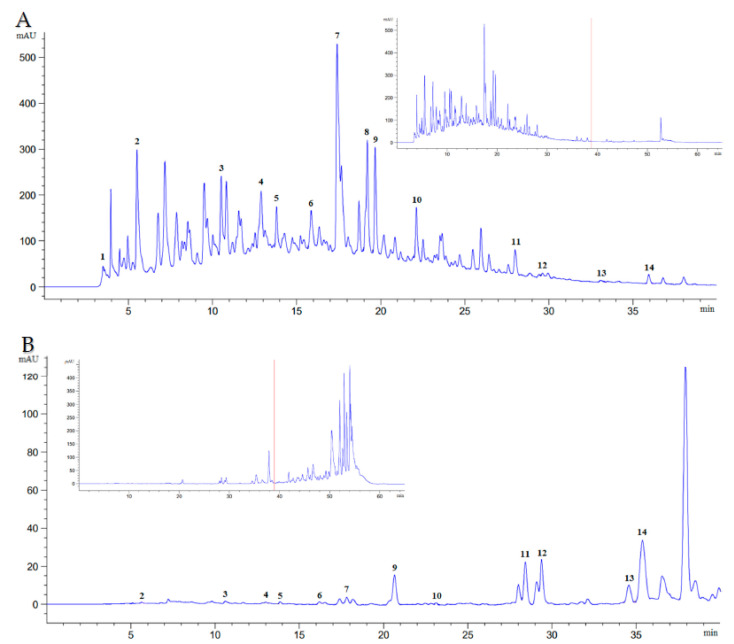
Chromatogram of Farang (*P. guajava* L.) leaves at λ = 272 nm, flow rate 0.9 mL/min, for 65 min. (**A**) methanolic extract (20× dilution); (**B**) hexanoic extract (20× dilution). Peak 1: ascorbic acid; 2: gallic acid; 3: catechin; 4: caffeic acid; 5: syringic acid; 6: rutin; 7: *p*-coumaric acid; 8: sinapic acid; 9: ferulic acid; 10: myricetin; 11: quercetin; 12: apigenin; 13: kaempferol; 14: eugenol.

**Figure 2 molecules-27-08987-f002:**
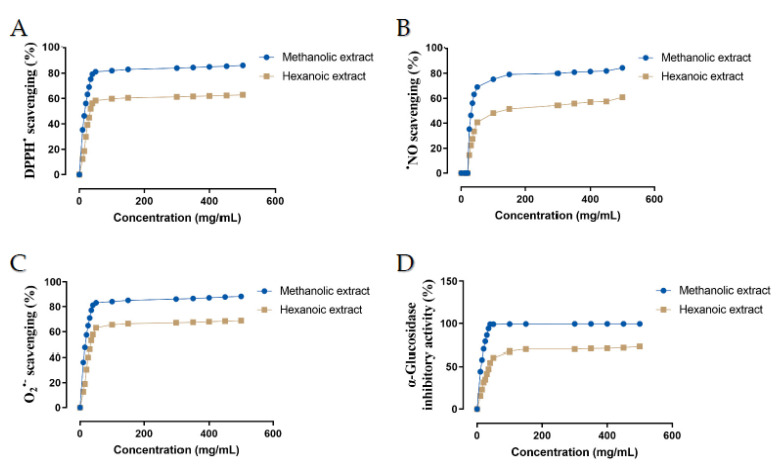
Effect of methanolic and hexanoic extracts of Farang (*P. guajava* L.) leaves against (**A**) DPPH^•^: 1,1-diphenyl-2-picrylhydrazyl; (**B**) O_2_^•−^: superoxide; (**C**) ^•^NO: nitric oxide; (**D**) α-glucosidase.

**Figure 3 molecules-27-08987-f003:**
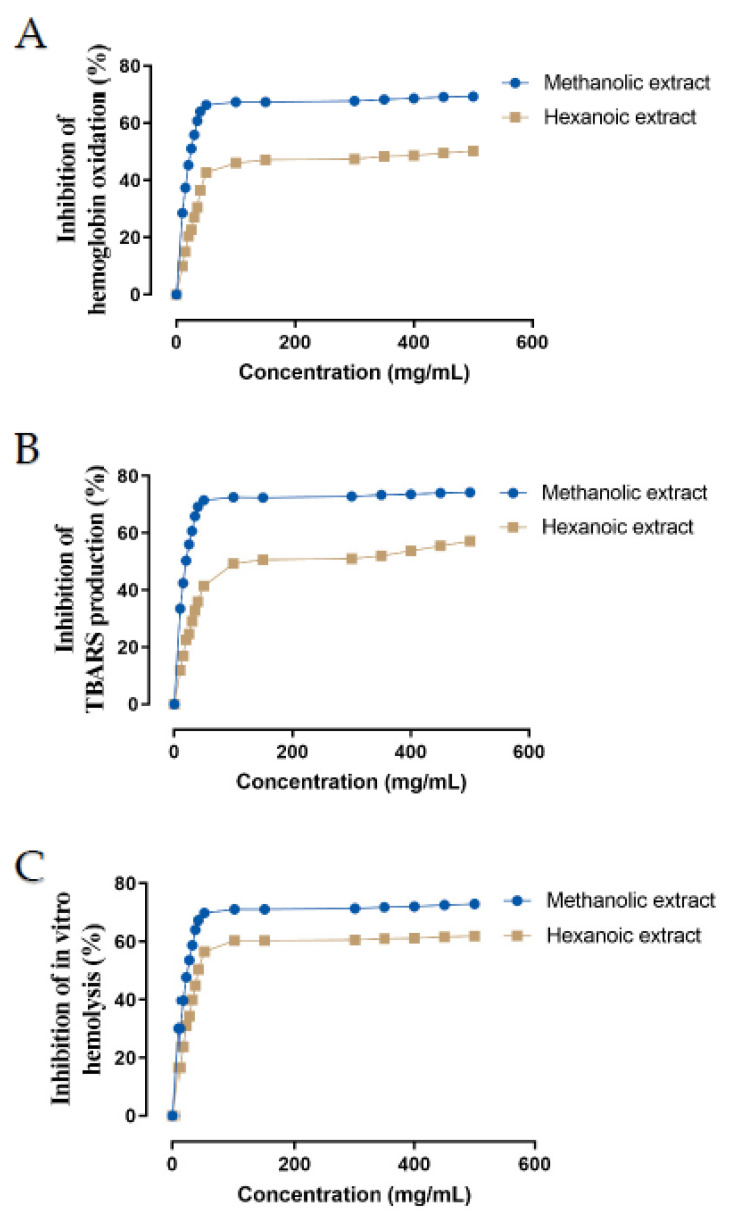
Effect of methanolic and hexanoic extracts of Farang (*P. guajava* L.) leaves against (**A**) hemoglobin oxidation, (**B**) lipid peroxidation, (**C**) hemolysis.

**Table 1 molecules-27-08987-t001:** Identifying and quantifying selected phytochemistry in methanolic and hexanoic extracts of Farang (*P. guajava* L.) leaves harvested from various cultivation sites.

Organic Compound	Wavelength Detection (nm)	Concentration (mg/g on Dry Weight Basis) ^a^		
		Methanol	Hexane	Average
CS1				
Ascorbic acid	272, 280, 310	0.18 ± 0.02	non-detectable	0.09 ± 0.01
Gallic acid	272, 280, 310	0.46 ± 0.10	0.01 ± 0.001	0.23 ± 0.05
Catechin	272, 280, 310	2.14 ± 0.11	0.01 ± 0.002	1.07 ± 0.06
Caffeic acid	272, 280, 310	0.47 ± 0.04	0.004 ± 0.001	0.24 ± 0.02
Syringic acid	272, 280, 310	0.15 ± 0.02	0.007 ± 0.002	0.08 ± 0.01
Rutin	272, 280, 310	0.34 ± 0.09	0.003 ± 0.001	0.17 ± 0.04
*P*-coumaric acid	272, 280, 310	0.90 ± 0.10	0.01 ± 0.001	0.45 ± 0.05
Sinapic acid	272, 280, 310	1.31 ± 0.22	non-detectable	0.65 ± 0.11
Ferulic acid	272, 280, 310	0.49 ± 0.10	0.020 ± 0.005	0.25 ±0.05
Myricetin	272, 280, 310	0.33 ± 0.07	0.01 ± 0.002	0.17 ± 0.03
Quercetin	272, 280, 310	0.17 ± 0.03	0.02 ± 0.002	0.10 ± 0.01
Apigenin	272, 280, 310	0.43 ± 0.001	0.23 ± 0.053	0.33 ± 0.03
Kaempferol	272, 280, 310	0.11 ± 0.01	0.04 ± 0.006	0.07 ± 0.01
Eugenol	272, 280, 310	0.26 ± 0.06	0.16 ± 0.22	0.21 ± 0.03
CS2				
Ascorbic acid	272, 280, 310	0.19 ± 0.05	non-detectable	0.09 ± 0.03
Gallic acid	272, 280, 310	0.51 ± 0.01	0.006 ± 0.001	0.26 ± 0.01
Catechin	272, 280, 310	2.13 ± 0.14	0.006 ± 0.002	1.07 ± 0.07
Caffeic acid	272, 280, 310	0.45 ± 0.01	0.005 ± 0.001	0.23 ± 0.01
Syringic acid	272, 280, 310	0.14 ± 0.01	0.004 ± 0.001	0.07 ± 0.003
Rutin	272, 280, 310	0.38 ± 0.01	0.002 ± 0.001	0.19 ± 0.01
*P*-coumaric acid	272, 280, 310	0.95 ± 0.07	0.01 ± 0.002	0.48 ± 0.04
Sinapic acid	272, 280, 310	1.41 ± 0.04	non-detectable	0.70 ± 0.02
Ferulic acid	272, 280, 310	0.52 ± 0.01	0.02 ± 0.004	0.27 ± 0.01
Myricetin	272, 280, 310	0.32 ± 0.01	0.01 ± 0.001	0.16 ± 0.005
Quercetin	272, 280, 310	0.18 ± 0.03	0.02 ± 0.01	0.10 ± 0.02
Apigenin	272, 280, 310	0.43 ± 0.01	0.26 ± 0.06	0.34 ± 0.03
Kaempferol	272, 280, 310	0.11 ± 0.03	0.04 ± 0.01	0.08 ± 0.01
Eugenol	272, 280, 310	0.25 ± 0.05	0.16 ± 0.03	0.20 ± 0.04

^a^ Data are reported as mean ± SD (*n* = 20 for each solvent). CS1: cultivation site I. CS2: cultivation site II.

**Table 2 molecules-27-08987-t002:** Total contents (mg/g on a dry weight basis) of tannin, flavonoid, cinnamic acid, essential oil, and vitamin in Farang (*P. guajava* L.) leaves harvested from various cultivation sites over a five-month period.

Organic Compound	CS1	CS2	SEM	*p* Value ^1^		
				Cultivated Site	Sampling Time	Interaction
Ascorbic acid	0.10	0.10	0.017	0.834	0.589	0.274
Gallic acid	0.36	0.34	0.105	0.887	0.980	0.343
Catechin	1.21	1.24	0.208	0.829	0.906	0.677
Caffeic acid	0.25	0.20	0.054	0.330	0.633	0.659
Syringic acid	0.08	0.07	0.012	0.271	0.129	0.609
Rutin	0.20	0.21	0.055	0.822	0.802	0.771
*P*-coumaric acid	0.69	0.62	0.112	0.432	0.364	0.136
Sinapic acid	0.78	0.87	0.126	0.328	0.585	0.130
Ferulic acid	0.29	0.29	0.072	0.972	0.794	0.846
Myricetin	0.24	0.24	0.057	0.962	0.492	0.592
Quercetin	0.12	0.10	0.023	0.394	0.061	0.229
Apigenin	0.52	0.44	0.134	0.390	0.997	0.949
Kaempferol	0.13	0.08	0.038	0.187	0.815	0.389
Eugenol	0.27	0.28	0.069	0.890	0.854	0.990
Total tannin	0.36	0.34	0.105	0.887	0.980	0.343
Total flavonoid	2.43	2.33	0.350	0.714	0.937	0.906
Total cinnamic acid	2.13	2.08	0.252	0.818	0.741	0.226
Total essential oil	0.27	0.28	0.069	0.890	0.854	0.990
Total vitamin	0.10	0.10	0.017	0.834	0.589	0.274

^1^*p*-Value: effect of cultivated site (CS1 versus CS2), effect of sampling time (August–December), and their interaction (cultivated site × sampling time); total tannin: sum of gallic acid; total flavonoid: sum of catechin, rutin, myricetin, quercetin, apigenin, and kaempferol; total cinnamic acid: sum of caffeic acid, syringic acid, *p*-coumaric acid, sinapic acid, and ferulic acid; total essential oil: sum of eugenol; total vitamin: sum of ascorbic acid.

**Table 3 molecules-27-08987-t003:** IC_50_ (mg/mL) values found in the antioxidant activity, α-glucosidase, hemoglobin oxidation, lipid peroxidation and hemolysis assays for Farang (*P. guajava* L.) leaves.

Item	Methanolic Extract	Hexanoic Extract
DPPH^•^	10.33 ± 0.008 ^a^	16.72 ± 0.007 ^b^
^•^NO	39.15 ± 0.019 ^a^	91.71 ± 0.169 ^b^
O_2_^•−^	10.35 ± 0.048 ^a^	18.72 ± 0.028 ^b^
α-Glucosidase	8.649 ± 0.020 ^a^	22.82 ± 0.016 ^b^
Hemoglobin oxidation	10.32 ± 0.030 ^a^	24.40 ± 0.884 ^b^
Lipid peroxidation	24.96 ± 0.023 ^a^	79.92 ± 0.126 ^b^
Hemolysis	10.26 ± 0.040 ^a^	18.14 ± 0.029 ^b^

Data are reported as mean ± SD (*n* = 6 for each assay); ^a,b^ values on the same row under each main effect with different superscript differ significantly (*p* < 0.05); DPPH^•^: 1,1-diphenyl-2-picrylhydrazyl; ^•^NO: nitric oxide; O_2_^•−^: superoxide.

## Data Availability

All data are contained within the article.

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
