# Peer review of "Farang (Psidium guajava L.) Dried Leaf Extracts: Phytochemical Profiles, Antioxidant, Anti-Diabetic, and Anti-Hemolytic Properties for Ruminant Health and Production"

_molecules, 2022, doi:10.3390/molecules27248987_

Round 1

Reviewer 1 Report

The manuscript “Farang (Psidium guajava) Dried Leaf Extracts: Phytochemical 2 Profiles, Antioxidant, Anti-Diabetic, and Anti-Hemolytic Prop- 3 erties for Ruminant Health and Production” is well written and discusses the phytochemical constituents, its associated benefits, its role in ruminant health. I have a few suggestions and it needs to be addressed before publishing.

Line 93: Vitamin C

Line 99: could be written as guava leaves instead of Guajava leaves

Line 106, 108: Please mention is it vitamin C or any other vitamins? This should be applied throughout the manuscript.

In table 1, nd non-detectable?  Please mention in words.

Title 2.4, Farang leaves (in plural)

Line 384 -385: I don’t understand this sentence. Was this extraction done using these solvents individually or followed one by one?

Line 396: was it one milligram?

Good luck with the submission. 

Author Response

Dear reviewer#1

We provide a point-by-point response to the reviewer’s comments as a word file, please see the attachment.

Best regards

Reviewer 2 Report

Please view the file attached 

Author Response

Dear reviewer#2

We provide a point-by-point response to the reviewer’s comments as a word file, please see the attachment.

Best regards
